# "Support for my dad would have benefited me because I was the one looking after him": A qualitative analysis of the support needs of young people exposed to Adverse Childhood Experiences

**Flo Avery** [1,2]*, **Michaela James**[1,2], **Laura Elizabeth Cowley** [2,3], **Lucy Griffiths** [2,3], **Mark A. Bellis** [4,5], **Karen Hughes**[5,6], **Sinead Brophy** [1,2]

**1** National Centre for Population Health and Wellbeing, School of Medicine, Swansea University, Sketty, Swansea, United Kingdom, **2** Population Data Science, School of Medicine, Swansea University, Sketty, Swansea, United Kingdom, **3** Environment and Health (ENVHE) Research Centre, Population Data Science, Swansea University, Sketty, United Kingdom, **4** Faculty of Health, Liverpool John Moores University, Liverpool, United Kingdom, **5** Policy and International Health Directorate, World Health Organization Collaborating Centre on Investment for Health and Well-Being, Public Health Wales, Wrexham, United Kingdom, **6** School of Health Sciences, Bangor University, Wrexham, United Kingdom

* f.l.avery.2226345@swansea.ac.uk

## Abstract

### Background

Adverse childhood experiences (ACEs) are associated with negative health and wellbeing outcomes. Ensuring young people receive timely and appropriate support after experiencing ACEs could improve these outcomes.

### Objective

This study aimed to explore what works to support young people living with ACEs; what support do they receive, and what are the characteristics of valuable help?

### Participants and Setting

Young people living in Wales aged 16–18 years (n = 559) completed an online survey about their ACEs and the help they did or did not receive with these experiences.

### Methods

Free text responses were analysed using reflexive thematic analysis. Public involvement workshops with young people were utilised to guide the analytic process.

### Results

Few participants reported accessing enough support. Five themes were developed: "Help me by helping my family"; "Talking to a trusted adult is helpful… until it's not"; "Being

**Data availability statement:** Data can be accessed from Open Science Framework. Unique, permanent digital object identifier: https://osf.io/2rcpf/.

**Funding:** This work was supported by the Economic and Social Research Council (ESRC) and Public Health Wales through their support of a PhD studentship. The Learned Society of Wales also provided support through a grant (38-WS-22). The funders had no role in study design, data collection and analysis, decision to publish, or preparation of the manuscript.

**Competing interests:** The authors have no conflicts of interest to declare.

informed: 'I was kept in the loop'", "Schools and colleges as sites of support" and "Loneliness and peer support".

## Conclusions

More support is needed for young people with ACEs. Young people find it helpful when their whole family is supported in times of adversity, not blamed. People who provide support should be empathic and non-judgmental. Young people would rather be spoken to about ACEs and 'kept in the loop' than have them treated as a taboo or sensitive subject. Experiencing ACEs can be lonely in the absence of peer support. Schools and colleges are acceptable sites of support and may be well placed to provide opportunities for peer support.

## 1. Introduction

Adverse Childhood Experiences (ACEs) are potentially traumatic events occurring before the age of 18 which broadly relate to child maltreatment or difficulties (e.g., substance use) within the family home [1]. It is estimated that around fifty percent of adults across Europe and North America have experienced at least one ACE during childhood [2]. Those who experience ACEs are at elevated risk of poor long term health outcomes [3,4]. Therefore, reducing the impact of ACEs is an important public health challenge. Addressing the impact of ACEs on mental health in the short term, e.g., in adolescence and early adulthood, may protect against worsening symptoms and prevent poorer outcomes in the long term [5,6].

An evidence gap concerning the type of support required by those affected by ACEs has previously been identified by Bellis et al. [7]. Lester et al. [8] reviewed qualitative evidence concerning the service needs of those affected by ACEs and concluded that provision of supportive relationships with adults was a key service need of young people affected by ACEs. Lorenc et al. [9] conducted a systematic review of systematic reviews and found that the strongest evidence for interventions on ACEs was for cognitive behavioural therapy (CBT), but identified significant gaps in the evidence, particularly concerning social and community-level interventions.

Much existing intervention research concerning ACEs and service needs predominantly includes young people already accessing support services because of their circumstances or experiences [8]. Many people with ACEs do not access services, and those that do may not disclose their experiences to a professional or even to others within their communities as ACEs are stigmatised [10–12]. As such, the needs of individuals with ACEs who choose not to disclose their experiences or are not able to access services are poorly reflected in existing studies. There is therefore a need to study young people's experiences of ACEs using a broader sample, thereby potentially capturing those not in receipt of help or support.

The aim of this study was to find out: what works to support young people living with ACEs? A survey was conducted which collected quantitative and qualitative data. This approach allowed for multiple participants, including those who have not disclosed ACEs and may therefore not be receiving help. Qualitative data from free text responses provided further insights into quantitative findings and allowed for a deeper exploration and understanding of young people's views.

## 2. Materials and methods

### 2.1 Design

This study uses a mixed methods approach including a phenomenological qualitative research approach with inductive thematic analysis. Centring the perspective of those with lived

experience is an appropriate way to gain insight into the sensitive phenomenon of growing up with ACEs [13]. Combining a survey approach with qualitative analysis techniques has allowed multiple perspectives to be captured without creating an untenably large volume of data for sensitive analysis.

## 2.2  Participants

Participants (n = 559) were adolescents aged 16–18 years living in Wales, recruited through social media (targeted Instagram adverts) and in-person on post-16 college campuses (through workshops in classrooms and stalls with flyers in common areas). Ethical approval was obtained from the Swansea University Medical School Research Ethics Sub-Committee (Reference 2022–0039).

## 2.3  Data collection

An online survey was developed and distributed using Qualtrics (https://www.qualtrics. com). All participants gave informed consent. The survey was completed by participants on their personal mobile devices. The survey captured sociodemographic characteristics, including age, gender, ethnicity, and whether participants were born in the UK. Participants were also asked whether they were in receipt of free school meals (FSM; a measure of poverty in the UK), whether they had experience of being in care, and characteristics of their household such as which other adults they lived with. Participants were asked about their exposure to 11 ACEs. The Behavioral Risk Factor Surveillance System list of experiences [14] was used. Participants were also asked if they had personally experienced a long-term health condition. This was suggested as a factor which may be relevant to ACEs in a focus group with young people with lived experience of ACEs and was therefore included for exploratory analysis.

   Participants who responded 'yes' to any of the ACEs questions or the question about long-term health conditions were immediately asked a follow up closed question about help (Did you feel you received enough help and support in this situation? (e.g., did you have someone to talk to about it, did you understand what help was available?); Response options yes, no, I don't know) and a follow up open-ended question (If you would like to provide details about what you found helpful, unhelpful, or help that you would have liked, please do so here). Responses were captured using free-text comments. All questions were optional. There was no direct compensation for taking part in the survey. A shopping voucher prize draw was conducted every two weeks. Participants who wanted to enter this prize draw provided their email address.

## 2.4  Quantitative analysis

Data were analysed using IBM SPSS Statistics (Version 20) and descriptive statistics were generated. ACEs were summed into a count variable.

## 2.5  Qualitative analysis

Free text responses to the open-ended question were analysed using reflexive thematic analysis (RTA) as outlined by Braun and Clarke [15,16]. Data were imported into NVivo and coded. All coding was conducted by the first author (FA), with co-authors MJ and SB providing regular sense-checking and peer review. FA as lead researcher was more immersed in data collection have spent time building partnerships with post-16 colleges across Wales and having previously worked with this age group as an educator. The authors therefore decided it would be appropriate for her to act as an active and reflexive lead researcher, immersed in

the research environment, whilst other authors provided supervision and review. No coding framework was used, instead codes were selected inductively from the data.

The data familiarisation phase began before data collection was complete and lasted several months from March 2023 to July 2023. Upon initial readings, each data item was treated equally in terms of importance. During this time, FA made notes on the data and reorganised data, reading extracts in different orders (e.g., chronologically by participant, or grouping extracts by ACE), reflecting on possible patterns arising from organising the data in different ways. Subsequently, initial codes were inputted as systematically and thoroughly as possible, even where this resulted in multiple codes with similar wording. Initial codes and possible groupings of codes were shared with two co-authors (MJ and SB) for sense-checking. From January 2024 to March 2024, a second round of data familiarisation following some time away from the project was conducted by FA. Initial codes were generated without reference to the first set of codes developed the previous year. Through a process of repetition and re-coding, higher-level codes were developed. Codes were organised and reorganised using NVivo and by manually printing codes and physically reorganising the data. As top-level codes emerged, data extracts included under each code were checked against the original participants' wider submission, to explore whether the phenomena captured by a code was representative of the wider dataset or specific to a sub-group of participants. Top-level codes were compared with possible groupings of codes from the previous year to reflect on similarities and where different interpretations had been constructed and under what influence. Over a process of recursive analysis, descriptive codes were developed into more complex themes which offer an interpretation of rich phenomena in the data.

Throughout this process, FA reflected regularly on the extent to which interpretations were grounded in data, or whether ideas from the literature, or from her time spent working with young people in schools and colleges were influencing the direction of theme development. Initial themes were shared with MJ and SB and reorganised and reworked as the analytic process continued.

Codes arising from Workshop A were compared with codes generated by FA and consideration given to any discrepancies in order to strengthen the validity of findings. Multiple interpretations were considered and refined, resulting in robust themes which capture distinct patterns of meaning and tell a story about the data [16]. Illustrative quotes were selected from the data; these have been presented with corrections to grammar and spelling. All authors peer reviewed the first draft of themes and gave feedback on these. KH and MB are internationally renowned in the field of ACEs and provided expert validations. Although codes were developed inductively, there was an element of structured analysis in theme development insofar as themes were organised to address the research question of what young people living with ACEs need, rather than a broader overview of what young people chose to say about ACEs.

## 2.6 Public involvement workshops

Public involvement workshops were conducted to ensure themes were relevant to young people living in Wales. Young people participated in these workshops and were given the opportunity to annotate and discuss the data, considering the meaning behind each response and linking it to their own personal experience of ACEs, or awareness of ACEs within their friendship group or wider community. For ethical reasons, participants were not required to disclose their lived experience of ACEs to researchers in order to take part in the workshop. Workshops were arranged through wellbeing lead staff members at the college, with whom FA had formed a relationship. These staff members were usually members of their college's safeguarding team so were able to suggest groups of young people to take part who had lived

experience of ACEs, but sensitive details were not shared. Participants were compensated for their contribution with shopping vouchers.

Workshops were conducted at two points. Workshop A was conducted in June 2023 in the data collection phase. It took place with two groups of 10–12 participants at a post-16 college which they were enrolled at. The aim was to involve young people in data analysis, specifically generating initial codes and searching for themes. Data from these workshops aided the reflexive process insofar as FA was able to compare codes generated by participants with her own coding process and reflect on any differences and where these may have arisen from. Participants completed discussion activities to introduce the concept of expertise through experience, and challenge the notion of academic knowledge being superior to knowledge through lived experience. Participants annotated and discussed printed survey comments, making direct recommendations for what they think young people need based on the comments, or writing further comments to extend and contextualise what they were reading in the survey comments based on their experiences. The first author FA conducted the workshops, she is a female PhD student with experience of teaching 11–18 year olds and working as a pastoral leader. Workshop participants were signposted to further support for those affected by ACEs and FA was also able to liaise with wellbeing staff and share workshop materials to ensure any safeguarding or wellbeing concerns arising could be addressed. A relationship with participants was not established prior to study commencement, however study participants had an existing relationship with their college's wellbeing lead who effectively acted as a gatekeeper in protecting participants' interests.

Workshop B was conducted in June 2024 towards the end of the data analysis phase, following the first draft of final themes. Seven participants attended a workshop at their post-16 college (a different college to Workshop A). Participants in Workshop B were all active users of the wellbeing provision offered by the college and all participants had a history of ACEs which college staff were aware of. Details were not shared with the researcher although most participants chose to disclose their ACEs history to the researcher during the workshop. Participants completed discussion activities to introduce the concept of expertise through experience. A plain-text summary of the five themes was displayed around the room, and participants were encouraged to annotate and discuss the themes and related recommendations and give their feedback on the findings. The objective of this workshop was to assess whether themes resonated with young people in Wales and sufficiently captured participants' experiences. Changes resulting from this workshop are detailed at the end of the results section.

## 3. Results

Between 2$^{nd}$ February and 10$^{th}$ October 2023, 889 survey responses were collected. Of these, 138 incomplete responses were removed (participants exited the survey early or provided an answer to two or fewer ACEs questions). A further 81 responses were removed because of poor quality responses (e.g., potentially bots). A further 111 responses were removed for the purposes of this analysis because they reported no ACEs. The final sample included 559 participants. Of these, 143 left at least one free text comment.

### 3.1  Sample demographics

Table 1 shows the sample demographics. Over two thirds (68.2%) were female (Table 1), 22.7% were male and 8.9% belonged to a gender diverse group. The majority reported white ethnicity (90.2%) and were born in the UK (96.1%). Participants were aged 16–18 years.

There was no major differences between male and female gender in number of ACEs reported overall. Female participants were slightly more likely to report living with someone with a mental health condition and sexual abuse. Those in the gender diverse group (who

**Table 1.** *Sample demographics from a survey of young people in Wales regarding their Adverse Childhood Experiences.*

| | | Number | Percentage |
|---|---|---|---|
| Gender | Male | 127 | 22.7 (51.9)* |
| | Female | 381 | 68.2 (48.1) |
| | Gender diverse | 50 | 8.9 |
| | Non-binary | 32 | 5.7 |
| | Other (free text option) | 18 | 3.2 |
| Age (years) | 16 | 191 | 34.2 |
| | 17 | 243 | 43.5 |
| | 18 | 122 | 21.8 |
| Ethnicity | White | 520 | 93.1 (92.5) |
| | Asian | 20 | 3.6 (3.0) |
| | Black | 6 | 1.1 (1.2) |
| | Mixed | 14 | 2.5 (1.4) |
| | Other (Arab or other group) | 6 | 1.1 (0.9) |
| UK born | Yes | 537 | 96.1 (97.8) |
| | No | 22 | 3.9 (2.2) |
| Region of Wales | Southeast Wales | 285 | 51.0 |
| | Southwest Wales | 143 | 25.6 |
| | Mid Wales | 23 | 4.1 |
| | North Wales | 103 | 18.4 |
| Number of ACEs | 1 | 125 | 22.4 |
| | 2–3 | 194 | 34.7 |
| | 4 or more | 240 | 42.9 |
| Health Condition | Yes | 265 | 47.4 |
| | No/Don't know/unanswered | 294 | 52.6 |

All questions were optional, and participants could select multiple answers to gender and ethnicity questions; totals do not sum to 100%.

*Numbers in brackets refer to equivalent nationally representative figures for Wales, released from the Secure Anonymised Information Linkage (SAIL) Databank.

selected non-binary gender or a free text option) reported ACEs at a higher rate than male and female participants; 62% of those in this group reported 4 or more ACEs compared with an average of 42.9% for the whole cohort.

### 3.2 Reporting of ACEs

The most reported ACE (53.0%) was sharing a household with someone who had a mental health condition, and the least reported ACE (11.1%) was physical neglect (Table 2).

### 3.3 Accessing help

Few participants reported accessing enough support with ACEs. Only 16.2% of those who had experienced sexual abuse reported receiving enough help (Table 3).

### 3.4 Qualitative analysis

A total of 325 comments were left. Of all participants, 143 (25.6%) participants left at least one comment. Supporting Information 1 shows the demographic characteristics of those who left

**Table 2.** *Individual Adverse Childhood Experiences reported in a survey of young people in Wales.*

| ACE reported | Number | % |
|---|---|---|
| Household mental health condition | 296 | 53.0 |
| Parental separation | 272 | 48.7 |
| Emotional neglect | 201 | 36.0 |
| Verbal abuse | 194 | 34.7 |
| Sexual abuse | 179 | 32.0 |
| Physical abuse | 177 | 31.7 |
| Household alcohol abuse | 147 | 26.3 |
| Household domestic abuse | 123 | 22.0 |
| Household drug use | 80 | 14.3 |
| Police or incarceration | 106 | 19.0 |
| Physical neglect | 62 | 11.1 |

**Table 3.** *Responses to 'Did you feel you received enough help and support in this situation?' regarding Adverse Childhood Experiences reported in a survey of young people in Wales.*

| ACE reported | Number reporting | Yes | % | Don't know | % | No | % |
|---|---|---|---|---|---|---|---|
| Sexual abuse | 179 | 29 | 16.2 | 25 | 14.0 | 124 | 69.2 |
| Verbal abuse* | 194 | 25 | 12.9 | 33 | 17.0 | 133 | 68.6 |
| Emotional neglect* | 201 | 30 | 14.9 | 33 | 16.4 | 134 | 66.7 |
| Household alcohol abuse | 147 | 31 | 21.1 | 29 | 19.7 | 85 | 57.8 |
| Household drug use | 80 | 19 | 23.8 | 14 | 17.5 | 45 | 56.3 |
| Physical abuse | 177 | 43 | 24.3 | 41 | 23.2 | 93 | 52.5 |
| Physical neglect | 62 | 11 | 17.7 | 17 | 27.4 | 32 | 51.6 |
| Household domestic abuse | 123 | 30 | 24.4 | 28 | 22.8 | 62 | 50.4 |
| Parental separation | 272 | 80 | 29.4 | 67 | 24.6 | 124 | 45.6 |
| Police or incarceration | 106 | 32 | 30.2 | 27 | 25.5 | 47 | 44.3 |
| Household mental health | 296 | 104 | 35.1 | 58 | 19.6 | 130 | 43.9 |
| Health condition* | 265 | 116 | 43.8 | 45 | 17.0 | 102 | 38.5 |

Note: All questions were optional; totals do not sum to 100%.

*Health condition is included here for comparison.

a comment, which were similar to the whole sample. Five themes (Table 4) relating to what helps young people with ACEs were interpreted from the dataset.

**3.4.1 Theme 1: Stigma and non-disclosure.** It was overwhelmingly clear from the survey responses that many young people did not tell anyone about their ACEs, or felt that although others were aware, their experience was never spoken about. This is an important barrier to accessing help, as young people cannot access help if they do not feel able to share or talk about their experiences. Overall, more participants commented on lack of support or poor support than positive support received. Very few participants commented on a positive experience of disclosure.

> *"I have never spoke about it" (Participant 446)*

> *"I didn't understand it was wrong at the time so I didn't tell anyone" (Participant 506)*

> *"I wish that it would have been a topic which people felt comfortable to speak to me about, instead of isolating my situation." (Participant 070)*

**Table 4. Major themes and sub-themes arising from a qualitative analysis of the support needs of young people in Wales with Adverse Childhood Experiences.**

| |
|---|
| Theme 1: Stigma and non-disclosure |
| 'I can't complain' |
| Being informed: 'I was kept in the loop' |
| Theme 2: Help me by helping my family |
| Multiple ACEs coexisting in a complex family system Family members provide support |
| Theme 3: Talking to a trusted adult is helpful… until it's not |
| Trusted, caring, empathic, confidential |
| Ignored, patronised, invisible |
| Theme 4: Schools and colleges as sites of support |
| Theme 5: Loneliness and peer support |
| Mental health and ACEs stigma overlap |
| Support groups and coping methods |

Many participants reported difficulty talking about their experiences, feeling alone, or having nobody to speak to. Some participants directly attributed their feelings to stigma, feeling silly, or feeling ashamed. Young people are aware of stigma around ACEs and this has the effect of making it more difficult for young people to talk about their experiences.

> *"Not many people know about it and it feels embarrassing to talk about it." (Participant 636)*

> *"I never knew where or how to speak to people about my feelings; I bottled it all up and kept them to myself." (Participant 557)*

Some young people did not seek help for their ACEs because they viewed these experiences as normal and therefore not something requiring help. The sub-theme of **"I can't complain"** was developed directly from a phrase in the data. Some participants explicitly stated that their experiences were normal or justified, for example that physical abuse was 'deserved'. Some experiences were constructed as context-dependent, for example verbal abuse being acceptable if it was during an argument, or not needing support for physical abuse if it 'only happened once'. Some participants recognised their own experiences as negative but identified others' perceptions of ACEs as 'normal' as a potential barrier to accessing help. A knowledge gap has therefore been identified concerning what ACEs are, their potential impact and how to access help for ACEs.

> *"I'm used to it, my mum is a functioning alcoholic. She drinks 3–4 crates of Carling a week but she gets up to work every day and pays the bills. I have a roof over my head and food I can't complain." (Participant 430)*

> *"I was hit as a child but not in a malicious hateful way." (Participant 167)*

**'Being kept in the loop'** In contrast to the majority of participants who did not receive informed help, a small number participants described 'being kept in the loop' as helpful for them in dealing with ACEs. Trusted adults who tried to ensure young people were informed about what was happening with their caregivers was experienced as reassuring. However, broadly, young people didn't feel well informed about their own experiences. Young people who were aware of their ACEs were often unsure of the details of what exactly had happened, when and how. Participants expressed uncertainty around whether they were entitled to help, or how to access help, and adults or other supportive people did not help to clarify this. The impact of stigma on young people's experiences is relevant here. Often stigma operates through an assumption that people do not want to talk about difficult things, but young

people feel this is more isolating and less supportive than being approached and spoken to about difficult topics, and being given opportunities to learn more.

> *"A whole group of family friends supported us and made sure my brothers and I understood what was happening and were kept in the loop about it all." (Participant 636)*

> *"Her being open about it and the resources available to me to be able to learn more about it [was helpful]." (Participant 415)*

There is a knowledge gap around ACEs which affects both young people and those who support them. Adults and peers who feel uninformed and uncomfortable talking about ACEs will not be able to ensure young people affected by ACEs are 'kept in the loop', and may prevent young people from identifying that they have been affected by ACEs. Similarly, young people who need signposting towards sources of help are often let down because those supporting them do not have the relevant knowledge. This contributes to young people feeling that they are struggling alone with ACEs.

> *"Well I wasn't really offered any support at all as I didn't realise it was an addiction until recently. My dad seemed normal and I never had reason to suspect otherwise." (Participant 123)*

> *"I was only 3, I never knew any different, but I never realised how much it hurt till I released not every family was like it" (Participant 544)*

> *"More knowledge/guidance as to when/what certain behaviours by adults become unacceptable [would help]." (Participant 537)*

### 3.4.2 Theme 2: Help me by helping my family.

This theme was interpreted from comments which emphasised that family members receiving appropriate support with their issues would directly benefit young people in the same household. Participants also expressed that family members should not be blamed for their experiences. Comments to this effect referenced a range of ACEs including having a household member with alcohol dependency, drug abuse and mental illness, and more broadly referenced domestic violence or experiencing poverty (which related to the young person experiencing neglect). Young people did not view themselves as victims of their family circumstances but instead experienced ACEs as complex events affecting the wider family in different ways.

> *"Support for my dad would have benefited me because I was the one looking after him which put a massive strain on me, my education and my mental and physical health… Addiction is a disease and it's not the addict's fault, people who struggle with addiction should get support because if they have kids or live with someone else, it's them that gets hurt the most." (Participant 549)*

This theme relates to a subtheme: **multiple ACEs coexisting in a complex family system**. This is a latent sub-theme insofar as it was interpreted from observing that in the data, ACEs were often referred to in a disorganised and overlapping manner, referencing concurrent ACEs in the same comment, or leaving comments in unexpected places. For example, the below comment, which mentions domestic abuse and parental separation, was left in response to a question about neglect:

> *"I was a baby and my mum had to get away from my abusive dad so we moved from [Town A] to [Town B] and we were homeless." (Participant 229)*

For many young people, ACEs are not experienced as discrete events, and young people do not think of ACEs in isolation. Instead, young people's experiences of ACEs are specific to, and difficult to separate from, their family makeup, which can make it harder to ask for help. The support which young people require does not necessarily mean support for one event or experience, but rather helping to make sense of wider events and patterns in a family which may not have a specific start or end point. This theme is also supported by comments which explicitly referenced 'not knowing' or confusion regarding the impact of ACEs.

> *"If you're used to it, or if you want to "defend" your family a person may be hesitant or reluctant to reach out for help, or downplay it. It can also be weird and take some effort to realise that the way one grows up is not necessarily the "normal" or "right" way." (Participant 358)*

> *"I had a little support… but it wasn't enough at all. I don't know what would have been helpful." (Participant 549)*

Many participants mentioned receiving help from a parent or family member, which supports the sub-theme of '**Family members provide support**'. This was the case even where young people received help from a family member who was directly involved in the ACE occurring, for example receiving support from a parent when parental separation was occurring, or support from a parent who was concurrently experiencing mental illness. The impact of stigma is also relevant here. Not all young people are supported by family members with the impact of their ACEs, but those who were appreciated open communication.

> *"My mum has PTSD and anxiety but she's an amazing mum and she's always there for me." (Participant 117)*

> *"My parents have tried to support me [during parental separation] in the best way they can and I know how much they love me." (Participant 310)*

**3.4.3  Theme 3: Talking to a trusted adult is helpful… until it's not.**  Many young people mentioned the importance of speaking to a trusted adult about their experiences. Who the adult is – a friend, a family member or a professional – is less important than their characteristics, including being perceived as trustworthy.

The subtheme of '**Trusted, caring, empathic, confidential**' refers to the characteristics which young people mentioned when talking about adults who helped them, or the help they wish they had received. These descriptors were sometimes applied to therapists or school staff. Public involvement workshop participants highlighted that support can be more effective when it comes from a known adult who embodies these characteristics and can offer personalised support to a young person who is struggling. Some participants mentioned that access to therapy and skilled mental health support embodying these characteristics was helpful. However, most spoke about trusted adults in general, in such a way that it was difficult to tell if they were referencing a counsellor, therapist, family member or other adult.

> *"To let me talk about what I had to go through without judgement and to keep it private." (Participant 053)*

> *"I would have liked to trust someone to tell." (Participant 153)*

> *"An alternative adult in my life who I could trust or rely on." (Participant 537)*

There were also some angry critiques directed specifically at therapists, particularly Child and Adolescent Mental Health Services (CAMHS) practitioners. The subtheme of '**Ignored,**

patronised, invisible' was developed from participant responses describing their frustrating interactions with professionals in fruitless attempts to access support. These experiences mirror the positive attributes referenced in the previous theme. This theme also links to comments about coping alone. Young people mentioned feeling that there is no-one available for them to talk to about their experiences, and feeling that if they initiated a conversation about their experiences it would not be welcomed or seen as appropriate. Participants referenced disclosing ACEs and then there being no follow up or acknowledgement or offer of help, which may make young people feel that sharing their experiences is not worth it. Public involvement workshop participants contextualised this finding by discussing the impact of austerity on under-resourced mental health services. This theme is illustrated both by those participants who had a negative experience with asking for help, and those who have not asked for help because they anticipated a negative experience.

> "CAMHS were aware and said they would help, a few weeks later they discharged me from therapy when I was still struggling and I was made to deal with it alone." (Participant 663)

> "I struggle to talk about feelings. Because of this the CAMHS people told my mum I refused to engage with them." (Participant 028)

> "I find it hard to talk about my experiences as it makes me feel shameful putting my problems onto other people." (Participant 292)

**3.4.4 Theme 4: Schools and colleges as sites of support.** For many participants, support received from schools and colleges was viewed positively. School staff, including teachers, teaching assistants and school-based counsellors, provided support in helping to understand ACEs and being a trusted adult who young people could speak to about their feelings.

> "One teacher gave me loads of support and helped me by talking through what was going through my head, educating me more on mental health and giving me coping techniques... This support was immense." (Participant 116)

> "I found the staff showing that they care about me by checking on me and doing stuff with me, it was helpful." (Participant 229)

School was also mentioned by participants who did not receive enough help with ACEs, and who desired further support within the school setting. Participants' experiences of pastoral support within school varied widely but even those that had had a negative experience discussed this in a way which implied that they felt school would have been an appropriate place to receive help. Young people found it unhelpful to be excluded from school when struggling with ACEs. A lack of follow up over absenteeism resulting from ACEs was also mentioned as unhelpful.

> "Both my dad and my brother have been to prison multiple times and I think being supported during school hours would have helped with this." (Participant 515)

> "I continually missed school and came in distressed and no teacher reached out to me." (Participant 202)

> "I had severe anxiety and the school I attend didn't help me to the extent that they should have." (Participant 421)

Participants mentioned wishing they had been 'noticed' at school as a way of receiving help, whilst others made recommendations that schools should do more, or that school staff

should receive more training to enable them to provide support. Again, this links to secrecy and stigma; many participants reported suffering in silence and discomfort with making disclosures.

> *"I just had no one, no support, no one to tell me how good I was dealing with things, no teachers would notice how depressed I was, no support workers would notice either." (Participant 549)*

> *"I never really spoke to anyone about how I felt and I wish I had had some more support in primary school" (Participant 521)*

**3.4.5  Theme 5: Loneliness and peer support.**  A number of participants talked about coping alone, and not receiving help because they did not tell anyone about their experience. This has been referenced in previous themes, but is also relevant in the context of the social situation and loneliness. Several participants directly referenced that confiding in peers was helpful.

> *"My friends did help a lot when I opened up to them." (Participant 638)*

> *"I didn't have anybody that I could turn to for help as I felt nobody actually cared what I was going through." (Participant 639)*

> *"My partner listened to me and validated my feelings." (Participant 303)*

The sub-theme **Mental health stigma and ACEs stigma overlap** was evidenced by participants who spoke about their mental health and their experiences in an indistinct manner. Several comments did not reference ACEs at all but instead focused on mental health. This was also evident in public involvement workshops, in which participants fluidly switched between referencing ACEs and mental health. This is important in the context of loneliness because both mental health and ACEs are stigmatised and so this presents two barriers to young people in accessing help.

> *"It's hard to ask for help when you don't know what's actually happening with your own mental health." (Participant 167)*

> *"I felt alone and depressed and no one ever helped me or tended to me or my other family members." (Participant 070)*

'**Support groups and coping methods**' was a sub-theme identified from views around social support and coping mechanisms, as well as schools as the site of support. Support groups encompass many of the recommendations contained within the data around peer support, being taught techniques, and combating stigma.

> *"I would have liked to know more about the experiences of others in similar situations to mine, and the struggles and coping mechanisms they had." (Participant 449)*

> *"I would have liked to [have] been given coping methods, communication skills and have the people who were supporting me to be more empathic." (Participant 549)*

## 3.5  Health conditions and ACEs

Alongside the ACEs questions, participants were also asked if they had personally experienced a long-term health condition, to explore whether this experience is relevant to young

people living with ACEs. Although living with a health condition does not sit within the ACEs framework of experiencing abuse or challenges within the family, 47% of participants indicated that they had experience of a long-term health condition, which is a high proportion. When asked about their health, many young people spoke about their experience of receiving support for mental health conditions, which relates to the overlapping nature of ACEs and wider concerns impacting mental health. Many young people affected by ACEs may be already living with the negative impact of those experiences on their mental health, and it is not necessarily helpful to differentiate between help for ACEs and help for a health condition where these factors are linked. Those living with physical health conditions and neurodivergent conditions (such as autism) also reported similar challenges to those which were discussed in the context of ACEs; difficulty in accessing support and experiencing stigma leading to exclusion and loneliness. However, young people's comments on their experience of living with a health condition did not mention non-disclosure or secrecy, in contrast to ACEs.

## 3.6  Public involvement workshops

In Workshop A, participants identified friends, family and peers as key sources of support. Participants discussed waiting lists and the importance of stigma and awareness-raising as a feature of help for young people, and the attributes of people who support young people. This input guided the initial coding process.

In Workshop B, young people gave their views on the first draft of themes. Regarding Theme 1, participants highlighted the possibility of peer support being a useful intervention for families, and the potential for family support to address the intergenerational nature of ACEs, for example by supporting parents with their recovery from childhood trauma. For Theme 2, participants emphasised the importance of receiving help from a known adult and discussed mental health services operating in the context of austerity and lack of resources. Participants agreed with Themes 3 and 4. The Theme 5 sub theme **Mental health and ACEs stigma overlap** was conceptualised following Workshop B. It had previously been identified as a middle-level code and referenced briefly in the context of overlapping ACEs in the family system, however workshop discussion identified this as an important aspect of experiences of loneliness, hence it was promoted to a subtheme.

## 4.  Discussion

The aim of this study was to explore what helps young people who are living with ACEs. A key finding was that young people living with ACEs report receiving inadequate levels of help and support with their experiences.

Qualitative findings suggest that young people want help for dealing with ACEs from their wider families, schools or colleges, and peers. However, many are not accessing support because they do not often feel comfortable discussing or even disclosing their experiences. Supportive responses to ACEs should support the whole family; for many young people, ACEs are not distinct experiences but rather features of a complicated family system. Young people also did not differentiate clearly between ACEs and their experience of mental health difficulties; for some, these experiences appear overlapping and indistinct. Young people whose caregivers experience domestic violence or alcohol abuse can be helped by ensuring their caregivers receive help directly. Family members who provide support to a young person should be supported to do so effectively. Empathic responses and opportunities to talk about feelings with trusted adults are important to young people. Young people want others to 'notice' when they are struggling to cope and provide help. Where young people have disclosed ACEs, it

is important for them and for those supporting them to be informed about ACEs and their impact.

### 4.1 Stigma and disclousre

Most perspectives from young people were phrased in terms of what they wished they had, rather than what help they did access. Lack of support can be related to stigma surrounding ACEs and relatedly to levels of non-disclosure. Young people who are burdened by ACEs are experiencing shame and loneliness in these experiences; it is better to talk about ACEs than treat them as a taboo or delicate subject. Reducing stigma around ACEs and ensuring young people feel emotionally safe enough to disclose their experiences is an important prerequisite to ensure that young people can benefit from the support that is available.

### 4.2 Family Support

Supporting the whole family is important for young people living with ACEs. This is consistent with previous research which has highlighted the importance of supporting individuals and the whole family [17]. Young people expressed that they do not necessarily blame their family members for their ACEs. For some, it is difficult to identify ACEs as events as opposed to background characteristics of their family environment. This is applicable to 'indirect' ACEs such as a family member who uses drugs or alcohol, and may not apply to experiences of child abuse. A key agency in the UK which supports a whole family approach is social care [18]. However there was no mention of social care at all from participants, either positive or negative. This could be because many participants were not 'known' to social care teams and therefore had had no contact, despite many participants reporting multiple ACEs. Lack of contact may also relate to fears around social care involvement and families being separated; this interpretation was mentioned by public involvement workshop participants.

Participants also discussed support from family members, both those who were and were not directly involved in ACEs. Previous research has been mixed; some studies identified positive parenting as a protective factor [19], but others found no association between parental relationships and ACE-related outcomes [20]. Support for parents and wider family members may be beneficial. Young people want someone to talk to who will keep them informed and who will not judge them for their experiences. Family members are equally as able to perform this supportive role for young people as professionals. Family peer support, in which adults with relevant lived experience provide peer support to parents, was mentioned as a possible intervention in public involvement workshops. This practice has been identified as potentially beneficial for parents supporting someone who is currently mentally unwell [21], and has been recommended for supporting parents with substance use disorder recovery and child welfare involvement [22]. This approach may be helpful for families living with ACEs.

### 4.3 Schools

Many young people reported receiving or wishing they had received support through school or college, possibly because they are already in contact with school and prefer to receive support from a known adult. Furthermore, accessing support in school could contribute to reduced stigma because it is somewhere young people already go. Not all participants had a positive experience of receiving help from school; some participants discussed wanting help from schools but not receiving it. There is a lack of evidence concerning trauma-based interventions suitable for non-clinicians such as school staff to deliver. Interventions based on cognitive-behavioural therapy (CBT) and those involving a caregiver have been identified as the most promising areas [23], which is consistent with the previous theme. Schools should

do more to provide ACEs support; as has been previously established in the literature, they are well located to provide this [24].

The help provided in schools should also be informed by the finding that stigma reduction and young people being informed about ACEs is an important feature of help. Peer support, coping skills, school and combating loneliness were all mentioned as important by participants. Peer support groups which are based in schools are suggested as a possible intervention which would have high acceptability to young people. Peer support has been suggested as a protective factor for those with ACEs [25], and "Peer support and healthy school climate", which appears on the Positive Childhood Experiences (PCE) scale, has also been suggested as protective [20]. Increased access to peer support may furthermore support those with ACEs by addressing loneliness. Availability of a school-based support group would directly address several of the recommendations and viewpoints expressed in the data. To support young people, schools should provide opportunities for peer support, through groups or otherwise, and ensure schools are experienced as safe environments.

### 4.4  Barriers

Not being listened to and not feeling able to disclose their experiences was a key barrier which prevented young people from accessing support. Young people want to be "noticed". Stigma continues to play a role here; professionals may avoid speaking about ACEs because they do not feel equipped to deal with difficult and sensitive topics. There is an ongoing policy discussion surrounding the potential usefulness of screening for ACEs. Finkelhor [26] has called for caution in promoting ACEs screening; it may not be considered acceptable to all groups to be asked about ACEs, and effective interventions for ACEs must be established for screening to be beneficial. However, literature from primary healthcare suggests that although those accessing care might experience discomfort being asked about ACEs, it can also be experienced as supportive if it is done sensitively [27]. Young people who have experienced ACEs and are currently experiencing poor mental health may find it harder to seek help. Stigma contributes to non-disclosure and secrecy for youth experiencing mental health problems [28], which may negatively influence help seeking for ACEs.

Normalisation is also a potential barrier; young people are not seeking help because they do not view their experiences as problematic, or are unsure whether others would view their experiences as deserving of help. Family loyalty may also play a role for some young people who feel they should be 'defending' their family. Normalisation has been identified as a factor preventing young people from disclosing and seeking help when they have experienced sexual abuse [29], and has also been suggested as a feature of how children understand domestic violence [30]. Universal interventions to educate young people about ACEs, what they are, and the potential of negative impact could help more young people to seek help. Waiting lists and a lack of signposting were also identified as barriers. This can be partly attributable to political austerity and underfunding of public services in a context of increasing burden of youth mental illness [31]. Nonetheless, professionals who encounter young people, including school staff, should be well informed about the impact of ACEs. This is particularly important for those who are likely to provide pastoral support.

### 4.5  Long-Term Health Conditions and ACEs

Disability and health conditions are relevant considerations for providing support for those affected by ACEs. In Wales, 17.8% of female 15–19 year olds and 14.8% of male 15–19 year olds live with some sort of health condition [32]. By contrast, in this survey 47% of respondents had this experience. It is also important to note, especially in the context of universal

mental health provision and early intervention, that the effects of ACEs on mental health may become apparent at a young age. Rates of diagnosis of a range of conditions such including autism, ADHD and anxiety disorders have all increased in recent decades [33]. Young people affected by ACEs are existing in the current context of increased awareness of neurodevelopmental and mental health conditions and may face challenges around accessing diagnosis which are unique to this generation. Support for ACEs should be neurodiversity affirming and care must be taken to ensure physical and mental healthcare are delivered in a complementary manner.

### 4.6  Gender and ACEs

Participants identifying as gender diverse (e.g., non-binary or chose to self-describe their gender) reported ACEs at a higher rate than male and female participants. This is consistent with existing research suggesting the LGBTQ + population experience higher rates of ACEs [34], although the reason for this trend has not been identified. 8.9% of all survey respondents were gender diverse, which is a large proportion compared to national census estimates. However, trans and non-binary people are more likely to be young compared with the average person [35], and furthermore the cohort responding to this survey are high in ACEs which might also explain high rates of response from gender diverse individuals. Gender diverse individuals may be particularly at risk of the negative effects of ACEs and this factor should be considered in the provision of support.

### 4.7  Strengths and Limitations

This study was the first to attempt to survey young people across Wales to understand their support needs. Utilising a public involvement approach was a particular strength in ensuring the study is relevant to young people. This survey was not nationally representative as it attracted a higher proportion of responses from those who had experienced multiple ACEs. However, a key purpose of the survey was to explore the support provided to young people living with ACEs, and thus this approach was successful in achieving a sample of young people with relevant insight and lived experience. Twenty-six percent of all participants left a comment, which cannot be considered representative of the whole sample. However, the demographic characteristics of those who left a comment is very similar to those who did not and therefore this should not have skewed responses (see Supporting Information 1). It is likely that some participants did not report all ACEs experienced, and therefore data may be incomplete; this is a potential limitation. A potential weakness of this study was that there was no qualitative follow up (e.g., focus groups) with the same participants to check interpretation of themes from potentially disparate responses. Although public involvement workshops have addressed this to an extent, these were one-off workshops and so the extent to which parity of power between young people and researchers could be reached was limited. Future research building on this study should seek to involve and embed youth perspectives through a comprehensive co-production approach, to strengthen findings.

### 4.8  Conclusion

Findings from this study can be used to improve the effectiveness of support available to young people who have experienced ACEs. A comprehensive understanding of the support needs of young people living with ACEs was achieved through the use of qualitative methods to gain the perspectives of those with lived experience of ACEs. Young people are affected by stigma and secrecy in navigating accessing help for ACEs, and it must be recognised that those who have multiple ACEs experience these in an overlapping manner and often in a way which

is indistinguishable from their wider family circumstances. Universal interventions should be considered to ensure adults who encounter young people affected by ACEs – either as professionals or part of a wider family or support system – are able to sensitively approach the topic in a non-judgemental manner with empathy and care, and effectively signpost to support. This includes challenging the notion of ACEs as 'normal', which may prevent young people accessing help. Young people with ACEs are supported where the whole family is supported and helped, not blamed. Schools and colleges have a pivotal role to play in the provision of ACEs support, both in the provision of supportive professionals who are known adults to young people but also as sites of peer support.

## Supporting information

**S1 Table. Sample demographics from a survey of young people in Wales regarding their Adverse Childhood Experiences, organised to show those who left a comment.**
(DOCX)

## Author contributions

**Conceptualization:** Flo Avery, Sinead Brophy.

**Data curation:** Flo Avery.

**Funding acquisition:** Flo Avery, Sinead Brophy.

**Methodology:** Flo Avery, Michaela James.

**Project administration:** Flo Avery.

**Resources:** Flo Avery.

**Supervision:** Karen Hughes, Sinead Brophy.

**Writing – original draft:** Flo Avery.

**Writing – review & editing:** Michaela James, Laura Elizabeth Cowley, Lucy Griffiths, Mark A. Bellis, Karen Hughes.

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
