## [Decision Letter · Decision Letter 0]

2 Dec 2024

PONE-D-24-46204“Support for my dad would have benefited me because I was the one looking after him”: A qualitative analysis of the support needs of young people exposed to Adverse Childhood ExperiencesPLOS ONE

Dear Dr. Avery,

Thank you for submitting your manuscript to PLOS ONE. After careful consideration, we feel that it has merit but does not fully meet PLOS ONE’s publication criteria as it currently stands. Therefore, we invite you to submit a revised version of the manuscript that addresses the points raised during the review process.

We look forward to receiving your revised manuscript.

Kind regards,

Anna Rachel Conolly, PhD, MSc, PG Dip, BA (hons)

Academic Editor

PLOS ONE

“This work was supported by the Economic and Social Research Council (ESRC) and Public Health Wales through their support of a PhD studentship. The Learned Society of Wales  also provided support through a grant (38-WS-22).”

“This work was supported by the Economic and Social Research Council (ESRC) and Public Health Wales through their support of a PhD studentship. The Learned Society of Wales  also provided support through a grant (38-WS-22).”

“This work was supported by the Economic and Social Research Council (ESRC) and Public Health Wales through their support of a PhD studentship. The Learned Society of Wales  also provided support through a grant (38-WS-22).”

Additional Editor Comments:

Dear Authors,

Thank you for submitting this interesting and pertinent piece of research. I believe that if you revise your article in-line with reviewer two's comments your work will become much stronger. I believe that that refining the manuscript’s structure and methodological clarity in the manner suggested by reviewer 2 will help the article reach its full potential and ensure it meets the highest academic standards.

I do hope that you will take the time to revise your article. You have addressed a critical and underexplored topic, and publishing your research would increase the knowledge base and enhance understanding and awareness of these issues.

Best wishes

A Conolly

Reviewers' comments:

Reviewer's Responses to Questions

**Comments to the Author**

1. Is the manuscript technically sound, and do the data support the conclusions?

Reviewer #1: Yes

Reviewer #2: Partly

2. Has the statistical analysis been performed appropriately and rigorously? 

Reviewer #1: N/A

Reviewer #2: N/A

3. Have the authors made all data underlying the findings in their manuscript fully available?

Reviewer #1: Yes

Reviewer #2: No

4. Is the manuscript presented in an intelligible fashion and written in standard English?

Reviewer #1: Yes

Reviewer #2: Yes

5. Review Comments to the Author

Reviewer #1: 1) The qualitative data from free text responses support the conclusions and this is a comprehensive paper with a methodology which allows the authors to capture some of the complexity of ACE experience from the perspective of those experiencing them, including those not currently accessing services.

3) I wasn't 100% sure how to respond to this since the authors state that while quantitative data are securely held in the Secure Anonymised Information Linkage Databank (SAIL Databank) at Swansea University, the qualitative data used to inform this study (from free text responses) is available upon request because SAIL is not optimised for qualitative data.

4). The manuscript is well-written and clearly presented with a high level of English.

Reviewer #2: The authors deserve commendation for addressing such a critical yet underreported topic. The experiences of children who endure Adverse Childhood Experiences (ACEs) represent an essential area of study that holds significant implications for both social and professional spheres. By shedding light on these issues, the researchers are making an invaluable contribution to a field that urgently requires further exploration and advocacy.

However, I have several concerns regarding the qualitative approach employed in this study, which raises doubts about the rigor and trustworthiness of the analysis. A key aspect of conducting qualitative research is clarity in the chosen paradigm that guides the study. The paradigm shapes the research aims, methodologies, and interpretative lens, and its absence here creates ambiguity for readers and reviewers alike. There appears to be a tendency toward "reversed engineering," where data collection precedes clear methodological planning. This lack of coherence undermines the rigor and trustworthiness of the study's findings, making it difficult for readers to follow the analysis process effectively.

The authors provide a detailed description of the qualitative analysis methods; however, it is concerning that only the first author (FA) was responsible for the coding process. While the co-production workshops described in the methods section may have aimed to enhance trustworthiness, their purpose and implementation are insufficiently detailed. Without a clear explanation of the workshop's objectives, it is unclear how they contribute to strengthening the validity of the findings. Moreover, the inclusion of inexperienced participants, such as "regular" adolescents, instead of individuals with lived experience relevant to ACEs, raises questions about their capacity to reflect meaningfully on the data.

Another significant concern is the lack of peer review or expert validation of the coding process, which is vital in ensuring the credibility of an inductive qualitative approach. The analysis of narrative data from 559 participants is a monumental task, and it is difficult to ascertain how one author could process such a large volume of data independently without jeopardizing the study's rigor. Furthermore, the relationship between workshops A and B remains unclear—are the participants the same, or are they distinct groups? This lack of clarity further complicates critical appraisal of the study design.

The study appears to align more closely with a phenomenological paradigm, as it aims to unveil the lived experiences and opinions of individuals who have endured ACEs. However, the authors need to clearly establish and articulate their paradigm to enhance the coherence and reliability of the study. If the authors choose to retain an inductive approach, they should explicitly address how they managed the extensive data set and enhanced trustworthiness, perhaps through techniques such as triangulation, reflexivity, or member checking. Alternatively, the authors might consider adopting a positivistic paradigm that facilitates a quantitative reporting style, which could be more suitable given the large data set.

To strengthen the study, I recommend acknowledging foundational works on qualitative research methodology, such as those published by Creswell, to better align the research design with its aims. Furthermore, I strongly suggest referencing exemplary qualitative studies, such as Shestiperov et al. (2024), which address a similar population with methodological rigor and reflective clarity. This publication offers a comprehensive framework for approaching analysis and trustworthiness techniques, and its reporting style serves as a model for presenting qualitative findings effectively by using SRQR Reporting checklist. The article is accessible here:

https://sigmapubs.onlinelibrary.wiley.com/doi/full/10.1111/jnu.12955

The SRQR Reporting checklist for qualitative study can be found here:

https://gh.bmj.com/content/bmjgh/5/11/e003433/DC1/embed/inline-supplementary-material-1.pdf?download=true

In conclusion, while the study addresses a highly significant and underexplored topic, substantial revisions are needed to enhance its methodological rigor and clarity. I strongly encourage the authors to refine their approach and incorporate the recommended strategies to ensure this important research can achieve its full potential in advancing mental health care.

6. PLOS authors have the option to publish the peer review history of their article (what does this mean? ). If published, this will include your full peer review and any attached files.

**Do you want your identity to be public for this peer review?** For information about this choice, including consent withdrawal, please see our Privacy Policy .

Reviewer #1: **Yes: ** Sarah Lester

Reviewer #2: No

---

## [Author Response · Author response to Decision Letter 0]

17 Jan 2025

Thank you for your useful feedback. Please see response letter attached.

---

## [Decision Letter · Decision Letter 1]

27 Feb 2025

“Support for my dad would have benefited me because I was the one looking after him”: A qualitative analysis of the support needs of young people exposed to Adverse Childhood Experiences

PONE-D-24-46204R1

Dear Dr. Avery,

We’re pleased to inform you that your manuscript has been judged scientifically suitable for publication and will be formally accepted for publication once it meets all outstanding technical requirements.

Kind regards,

Emily Lund

Academic Editor

PLOS ONE

Additional Editor Comments (optional):

Reviewers' comments:

Reviewer's Responses to Questions

**Comments to the Author**

1. If the authors have adequately addressed your comments raised in a previous round of review and you feel that this manuscript is now acceptable for publication, you may indicate that here to bypass the “Comments to the Author” section, enter your conflict of interest statement in the “Confidential to Editor” section, and submit your "Accept" recommendation.

Reviewer #1: (No Response)

Reviewer #2: All comments have been addressed

2. Is the manuscript technically sound, and do the data support the conclusions?

Reviewer #1: Yes

Reviewer #2: Yes

3. Has the statistical analysis been performed appropriately and rigorously? 

Reviewer #1: N/A

Reviewer #2: Yes

4. Have the authors made all data underlying the findings in their manuscript fully available?

Reviewer #1: No

Reviewer #2: Yes

5. Is the manuscript presented in an intelligible fashion and written in standard English?

Reviewer #1: Yes

Reviewer #2: Yes

6. Review Comments to the Author

Reviewer #1: I answered no to question 4 as there were certain restrictions which prevented the data from being made fully, publicly available which the authors have adequately explained.

Reviewer #2: Amazing work! Written well and accomplished all that was needed. Just one technical point needs to be addressed. Even though you mentioned that you added that quote, I didn’t see that you included article No. 13 in your bibliography list. Please update and add it so it will be adequate to the writings.

After In my opinion, this work is so important and deserves to be published.

Here is the APA citation for the missing article:

Shestiperov, A., Grinstein-Cohen, O., Lindell, D., Irani, E., & Kagan, I. (2024). Lived experiences: Growing up with a seriously mentally ill parent. Journal of Nursing Scholarship: An Official Publication of Sigma Theta Tau International Honor Society of Nursing, 56(3), 357–370. https://doi.org/10.1111/jnu.12955

7. PLOS authors have the option to publish the peer review history of their article (what does this mean? ). If published, this will include your full peer review and any attached files.

**Do you want your identity to be public for this peer review?** For information about this choice, including consent withdrawal, please see our Privacy Policy .

Reviewer #1: **Yes: ** Sarah Lester

Reviewer #2: No

---

## [Editor Report · Acceptance letter]

PONE-D-24-46204R1

PLOS ONE

Dear Dr. Avery,

I'm pleased to inform you that your manuscript has been deemed suitable for publication in PLOS ONE. Congratulations! Your manuscript is now being handed over to our production team.

Kind regards,

on behalf of

Dr. Emily Lund

Academic Editor

PLOS ONE